# Phenotypic Characteristics Associated with Shelter Dog Adoption in the United States

**DOI:** 10.3390/ani10111959

**Published:** 2020-10-24

**Authors:** Cassie J. Cain, Kimberly A. Woodruff, David R. Smith

**Affiliations:** 1Department of Pathobiology and Population Medicine, Mississippi State University College of Veterinary Medicine, Mississippi State, MS 39762, USA; cjc595@msstate.edu (C.J.C.); DSmith@cvm.msstate.edu (D.R.S.); 2Department of Clinical Sciences, Mississippi State University College of Veterinary Medicine, Mississippi State, MS 39762, USA

**Keywords:** animal shelter, shelter dogs, adoption, adopter preference, United States

## Abstract

**Simple Summary:**

United States animal shelters care for unwanted dogs until they are adopted, transferred to another facility, or euthanized. Certain characteristics have been previously studied to determine the adoptability of shelter dogs possessing such traits. However, previous studies are typically limited in sample size, shelter geographic location, and/or the number of shelters participating in the study; these reduce the generalizability of the results. To better understand predicters of shelter dog adoptability, the aim of this study was to identify dog characteristics predictive of adoption. This study helps us understand adopters’ preferences for certain shelter dogs, which may be useful to help shelters increase adoption rates and ultimately reduce shelter dog euthanasia.

**Abstract:**

The objective of this study was to identify phenotypic characteristics of dogs predictive of adoption after being received into a shelter. Individual dog records for 2017 were requested from shelters in five states that received municipal funding and utilized electronic record keeping methods. Records from 17 shelters were merged into a dataset of 19,514 potentially adoptable dogs. A simple random sample of 4500 dogs was used for modelling. Variables describing coat length, estimated adult size, and skull type were imputed from breed phenotype. A Cox proportional hazard model with a random effect of shelter was developed for the outcome of adoption using manual forward variable selection. Significance for model inclusion was set at alpha = 0.05. Dogs from shelters in the North were more likely to be adopted than dogs from shelters in the South (hazard ratio (HR) = 3.13, 95% C.I. 1.27–7.67), as were dogs from Western shelters versus those from Southern shelters (HR = 3.81, 95% C.I. 1.43–10.14). The effect of estimated adult size, skull type, and age group on adoption were each modified by time in the shelter (*p* < 0.001). The results of this study indicate that what dogs look like is predictive of their hazard for adoption from shelters, but the effect of some characteristics on hazard for adoption depend on time in the shelter. Further, this study demonstrates that adopters prefer a certain phenotype of shelter dog including those that are puppies, small sized and not brachycephalic, when accounting for time in the sheltering environment.

## 1. Introduction

In recent years, the United States dog sheltering industry has experienced a decline in shelter dog euthanasia and an increase in favorable shelter dog outcomes including adoption, transfer, and return to owner, collectively known as “live release” [1]. This decline in euthanasia may be due to population control through spay/neuter programs, the increasing availability of affordable pet veterinary care, or the rise of adoption-guarantee shelters [2]. A similar decline can also be seen in shelter dog populations in Europe. However, as many European countries have strict mandates and legislation in place to help reduce shelter dog populations [3,4,5], such enforcement may be the explanation for euthanasia decline. The United States does not currently have any national legislation regarding managing dogs without owners, as such legislation is optionally mandated by individual states. This absence may contribute to US pet overpopulation and shelter pet euthanasia.

It has been estimated that only 30% of US dog owning households obtain their dogs from a shelter [6]. This implies that, when dogs enter the shelter, they are in competition with one another for permanent homes and shelter space. One solution to lessen shelter overcrowding and re-home dogs more efficiently would be for shelter personnel to identify specific phenotypic traits that adopters prefer and allocate resources to the adoption of such dogs. Such phenotype identification may help improve shelter live release rates and ultimately lower shelter dog euthanasia.

The results of previous studies suggest that phenotypes such as age, coat length, color, size, and breed are associated with dog outcome. For example, puppies had lower rates of euthanasia [7] and greater chances of adoption [8]. Lighter colored dogs were more likely to be adopted than their darker counterparts [9], but medium and long coated dogs in the United Kingdom were more likely to be adopted, as well as small dogs and purebreds [10].

Another factor commonly associated with dog outcome has been breed. For example, in one study, breed was more predictive of adoption than coat color [11]. Others have found that certain breed groups are more adoptable or have shorter lengths of shelter stay [12,13]. However, shelter employee visual identification of dog breed is often not consistent with genetic breed analysis [14] and only 5% of dogs entering shelters are purebred [15]. The misclassification of breed among shelter dogs makes it difficult to study the effect of breed on shelter outcomes.

Studies focused on adopter preference suggest that dog appearance is one of the most important factors considered prior to adopting shelter dogs [16,17,18,19]. Understanding adopter preferences could help shelter employees make evidence-based decisions regarding dogs’ potential risk for adoption as they are admitted into the shelter’s care. However, our current ability to quantify this risk is limited because of small numbers of shelters, small numbers of dogs, or limited geographic range. Therefore, the objective of this study was to identify the phenotypic traits of shelter dogs associated with risk for adoption using a larger number of dogs coming from representative regions of the United States and considering length of stay in the shelter.

## 2. Materials and Methods

Shelters were chosen for inclusion in this study from a previously compiled census of shelters in five study states: Mississippi, Pennsylvania, Michigan, Colorado, and Oklahoma. This list represented 342 shelters. Only 86 shelters that received municipal funding and kept electronic records were included in the final shelter frame because municipally funded shelters were those shelters most likely to be open-admission and electronic records were necessary to facilitate data collection. Shelter contact was attempted twice by email, in which either a copy of the shelter’s records from 1 January 2017–31 December 2017, or the login information to their commercial record keeping database was requested. Access to commercial record keeping databases did not include access to confidential shelter documentation or information. After two contact attempts, no further contact was sought. Seventeen of 86 (20%) shelters provided records for this study.

For some shelters, records included duplicate entries of dogs. Such dogs were identified by the shelter’s database-derived animal identification number that was associated with each dog upon intake. When possible, duplicate dogs were merged into single entries in the final dataset. Length of Stay (LOS) was then calculated as the number of days between intake and the first outcome.

The subjects of interest in this study were dogs with the potential for adoption. Therefore, dogs with an outcome of owner requested euthanasia, returned to owner, or deceased on arrival were not included in the final dataset. Dogs with a LOS equal to zero days were also excluded from analysis. The final dataset contained 19,514 unique dogs.

Dogs may have had more than one outcome if they were adopted then returned to the shelter post-adoption several times, or if they were admitted more than once to the shelter as a stray. Approximately 1541 dogs had more than one outcome and LOS which was accounted for by creating multiple outcome and LOS variables. No dog had more than four outcomes. Only the first outcome and first LOS were evaluated in the analysis.

### 2.1. Dataset Variables

Because of the common misidentification of dog breed, phenotypic traits were imputed from each listed breed, including predicted adult size, coat length, and skull type. These variables were created by searching public pet adoption websites to visually standardize the sheltering industry label of each breed. For instance, if a dog in the dataset was identified as a husky, the term “husky” was entered in the search platform, and current shelter dogs identified as huskies were examined to standardize the appearance of their skull type, coat length, and estimated adult size. In this example, dogs identified as a husky would be classified as mesocephalic, long coated, and medium sized. Although this imputed information provides additional phenotypes for each shelter dog in the dataset, it is noted by the authors that if breed were a more reliable variable, it would be the preferred variable for analysis rather than imputed breed variables. Correct breed classification offers additional information that is lost when imputing breed, such as behavioral traits and other phenotypic indicators that may help predict shelter dog outcome. The authors understand that if some breeds were misidentified prior to phenotype assignment, then it is possible that the frequencies of imputed phenotypes are falsely elevated or reduced depending on how the breed was identified in the records provided. However, because such misclassification exists between shelters, the authors support the decision to impute phenotypes from breed.

Dogs were considered to have a brachycephalic skull type if their standardized breed phenotype had shorter heads and wider skulls (e.g., pugs, bulldogs). Dolichocephalic skulled dogs had longer heads, with characteristic long noses (e.g., hounds, collies). Dogs categorized as mesocephalic skull type had heads that were a fair medium between the two extremes (e.g., labradors, cocker spaniels). Guidelines describing skull classifications in more detail were used to aid in assigning skull types to each breed which allowed for decreased dog misclassification [20,21]. These guidelines classify each breed using specific skull measurements such as cephalic index and eye distance.

Breeds were categorized by body weight as “small”, “medium”, “large”, and “giant” if their expected adult weights were less than or equal to 13.6 kg, greater than 13.6 kg to less than or equal to 22.7 kg, greater than 22.7 kg to less than or equal to 31.8 kg, or greater than 31.8 kg, respectively. If dog records had sizes reported, then these reported sizes remained and were analyzed further. However, if dogs were puppies, then the size entry was changed to reflect breed estimated adult size. Because giant dogs only represented 2% of entries in the dataset, the giant category was combined with the large category. Outlier dog weights were removed from the dataset. Outlier weights included one 503-kg dog and twelve 0-kg dogs. The coat length variable was assigned as either short, medium, or long. Because medium coated dogs included only 10% of entries, the medium coat length group was combined with the long coat length group.

A blockhead variable was imputed from primary and secondary breeds to identify dogs that characteristically have square shaped heads. If dogs were described as pit bulls, Staffordshire terriers, boxers, Cane Corsos, mastiffs, English Bulldogs, bulldogs, American Bulldogs, or rottweilers, they were considered to be blockheaded [22].

Primary and secondary coat colors were categorized into 8 colors (black, brown, red, grey, white, tan, yellow, and blue) from 44 different variants of color reported in the provided dog records.

Age group was categorized as “puppies,” “young adults,” “adults,” and “seniors” if the reported age was less than or equal to 6 months, greater than 6 months to 2 years, greater than 2 years to less than 8 years, or greater than or equal to 8 years, respectively. Because there is no phenotypic indicator to differentiate the age of young adults versus adults, as there is for puppies at approximately 6 months with the eruption of permanent canines, the young adult group was combined with the adult group.

The region where each shelter was geographically located was categorized as southern, northern, or western. The southern region included shelters from Mississippi and Oklahoma. The northern region included shelters from Michigan and Pennsylvania. The western region included shelters from Colorado.

### 2.2. Data Analysis

Inferential statistics were computed using SAS for Windows v9.4 (SAS Institute, Inc., Cary, NC, USA), and sample size calculations were performed using Epi Info (CDC, Atlanta, GA, USA). Crude descriptive statistics were completed using spreadsheet software (Excel v16, Microsoft, Redmond, WA, USA).

An extended Cox proportional hazard regression model was created through manual forward variable selection. Variables were retained in the model if Wald type 3 *p*-values were significant at an alpha equal to 0.05. Shelter was included as a random effect in the model. Age group, coat length, estimated adult size, skull type, presence of a blockhead, region, primary coat color, and gender were tested for inclusion as fixed effects. To improve model stability, the length of stay was limited to 80 days, after which a dog was considered censored.

To reduce the ability to detect very small differences in independent variables, a simple random sample was taken from the dataset of 19,514 dogs using SAS, PROC SURVEY SELECT. Using the cohort study sample size calculator from Epi Info, a sample size of approximately 4500 dogs was determined to be sufficient to detect a risk ratio of 0.88 at a 95% confidence level, assuming 95% power.

The proportional hazard assumption was tested by creating and testing a time interaction variable for each fixed effect. Variables with time interactions were depicted graphically using methods described by Dohoo [23]. Multiple comparisons were adjusted using Tukey–Kramer methods. Dogs with incomplete information for all of the phenotypes included in the model were ultimately excluded.

## 3. Results

Of the 342 shelters included on the shelter list, 86 (25%) shelters met both criteria of receiving municipal funding and keeping electronic records. Seventeen of 86 (20%) shelters provided complete records for analysis. Of the 19,514 dogs with the potential to be adopted, 12,793 (66%) were adopted, 2657 (14%) were euthanized, 3141 (16%) were transferred, 153 (<1%) either escaped, were missing, or died in the shelter, and 770 (4%) had unknown outcomes. Of the 19,514 dog data set, 509 (3%) dogs were censored at 80 days, as they had not yet experienced an outcome by that time. These censored dogs are accounted for in the 770 dogs with an unknown outcome. The median LOS for dogs to be adopted was 10 days with a mean (standard deviation) of 14.1 days (13.4 days). Using the dataset of 19,514 dogs and the random sample of 4500 dogs, the frequency of variables tested for model inclusion are outlined in Table 1. From the random sample of 4500 dogs used for model building, 2988 (66%) were adopted. Every shelter was included in the sample.

Presence of a blockhead and skull type were collinear when both variables were included in the model. Therefore, skull type was chosen for model inclusion as it provided better model fit. Variables in the final model included region, age group, predicted adult size, and skull type (Table 2).

Region was associated with adoption and met the proportionality assumption. Dogs in the north had adjusted hazard ratios (HR) for adoption different from those in the south (HR = 3.13, 95% C.I. 1.27–7.67) as well as dogs in the west compared to those in the south (HR = 3.81, 95% C.I. 1.43–10.14). Adoption of dogs from the north was not different from adoption of dogs from the west (*p* = 0.845).

Age group had an interaction with time (*p* < 0.001) and did not meet the proportionality assumption. Hazard ratios for puppies and seniors compared to adult dogs were graphed over the LOS in days (Figure 1). A solid line is graphed at HR = 1 used to demonstrate “null” HR value.

Predicted adult size had an interaction with time (*p* = 0.0005) and did not meet the proportionality assumption. Hazard ratios for small and medium size dogs compared to large dogs were graphed over the LOS in days (Figure 2). A solid line is graphed at HR = 1 used to demonstrate “null” HR value.

Skull type also had an interaction with time (*p* < 0.001) and did not meet the proportionality assumption. Hazard ratios for brachycephalic and dolichocephalic dogs compared to mesocephalic dogs were graphed over the LOS in days (Figure 3). The hazard of adoption of dolichocephalic dogs compared to mesocephalic dogs was not different over time (*p* = 0.473). A solid line is graphed at HR = 1 used to demonstrate “null” HR value.

## 4. Discussion

To our knowledge, a multiregional study of this scale has not previously been conducted and thus the results of this study may be more representative of the shelters across the US than previous studies. An important finding of this study was that as LOS increased, the hazard of adoption changed for phenotypes of age group, predicted adult size, and skull type. It was through the use of survival analysis methods that we were able to detect these relationships. The methods of analysis used in previous studies would not have detected these time-dependent interactions.

When comparing puppies to adults, we found that puppies had a greater hazard for adoption that generally decreased as LOS increased. Although the hazard of adoption changed for puppies over time, puppies were found to always have a higher hazard for adoption than adults. This finding demonstrates that the longer puppies stay in the shelter, the lower the chance that they will be adopted. Previous studies have reported that, as dogs increase in age, their chance for adoption decreases, thus making puppies the most adoptable age group [8,16,24,25,26]. It has also been previously reported that puppies have the shortest length of shelter stay [12,13,22].These results may be explained by a concept known as “baby schema” which could be used to help describe the general attractiveness and preference for younger individual’s features. Previous studies have identified that both adults and infant humans have a preference for puppies opposed to adult dogs when visually assessing their cuteness [27]. Although previous literature does identify that puppies are more adoptable than adults, this study enhances what is known about puppy adoptability by identifying a LOS relationship that was previously unknown.

When comparing seniors to adults, seniors had a lower hazard for adoption which steadily increased as LOS increased. At approximately 50 days, the adoption hazard ratio (HR) of seniors to adults was equal to 1. This HR demonstrates that after 50 days, seniors are considered more adoptable than adults within the study time frame of 80 days. This relationship, to the authors’ knowledge, has not been previously described. One study found that dogs 8 years or older are “marginally” more likely to be adopted compared to 5–7 year old dogs [10]. Another study found that when only analyzing geriatric dogs, their average LOS was 89 days, but that health condition was the most predictive factor for adoption [28]. Our findings indicate that the longer senior dogs remain in the shelter, the more adoptable they become relative to other age groups with the same LOS. We speculate that this may happen because some adopters or rescue groups enter shelters specifically to adopt older dogs. Some rescues will only adopt geriatric dogs from shelters for end of life care. Another explanation may be that shelters increasingly promote or advertise the adoption of senior dogs once they have been available for adoption for a certain amount of time. However, this study did not collect shelter dog promotional information, so these are only speculations. Identifying this senior-time-adoptability relationship is important because it demonstrates that the longer seniors are in the shelter the more adoptable they become relative to other age groups.

We also found that the effect of dog size on adoptability changes as LOS increases. Both small dogs and medium dogs initially have a greater HR for adoption compared to large dogs, which gradually decreases over time. A size-outcome relationship has been identified in previous studies. For instance, one study states that small dogs have the shortest LOS, but medium dogs have the longest [12]. Another study found that small dogs were more likely to be adopted than large dogs [9]. However, these studies do not identify a change in risk of adoption as LOS increases. Quantifying a specific time period in which small- and medium-sized shelter dogs have significantly less adoptability is a novel, useful finding for shelter employees. Shelters may be able to use this information to promote adoption of these dogs during their initial days upon intake, as they become less adoptable the longer they remain in the shelter. An initial preference for small- and medium-sized dogs, as opposed to large dogs may be explained by weight restrictions for adopters’ living conditions such as rental properties, or the general ease of care for dogs that are smaller in size.

The final time-dependent phenotype identified in this study was skull type. This is the first study to the authors’ knowledge to analyze skull type as a predictor for adoption and the first study to identify a time-dependent relationship between this phenotype and adoption. Previous work found that dogs with a blockhead had lower odds of live release [22]. However, this previous study failed to account for additional skull types and the LOS effect on adoption risk. The finding that brachycephalic dogs are always less adoptable than mesocephalic dogs may be explained by the brachycephalic categorization of breed. Some of these brachycephalic breeds include pit bulls, bulldogs, and rottweilers, which are often overrepresented in US shelters, are subject to breed mislabeling [29], and are often targeted in breed specific legislation [30]. However, because breed identification is highly variable among shelters, further analysis to understand if these specific breeds are causing the brachycephalic effect on adoptability is not possible with the data provided. Another possible explanation indicating why brachycephalic dogs are less adoptable, may be because some small-sized, brachycephalic dogs are more prone to chronic respiratory problems, such as Brachycephalic Obstructive Airway Syndrome [31]. The finding that the length of time brachycephalic dogs are in the shelter does not change their overall adoptability should allow shelter staff to make more efficient and objective decisions regarding brachycephalic dogs’ outcomes. It should be noted that additional specific facial characteristics may also affect dog outcome, but this study does not have the resolution to determine such characteristics on an individual level.

The only static relationship identified in this study was that of region. Dogs from the southern region were significantly less likely to be adopted compared to dogs in the northern and western regions. This result agrees with previous findings [32]. States in the southern United States tend to have a larger population of homeless animals which may be explained by mild winter temperatures, conducive for stray animals to survive. Furthermore, southern states tend to have less stringent dog legislation enforcement, thus allowing for a greater number of stray dogs to enter shelters year-round and reducing individual shelter dogs’ chances of adoption. This is opposed to many northern states which typically have harsh winters and enforce stricter pet ownership laws including leash laws or spay-neuter requirements, thus lowering the regional shelter dog population [33]. Regions were used for geographic categorization based on the sampling of states from a previous study. Additional socio-economic information may also influence why shelters in certain regions adopt out more dogs, but because of the variation between states in each region, it is impractical to generalize such information for a region when uniformity does not exist among states. Some regions, such as the western region, contained one state: Colorado. The results from this study may be directly applicable to shelters within Colorado, but not necessarily to all other states in the Western US. This is also the first study to the authors’ knowledge, which analyzes factors affecting shelter dog adoption in Mississippi, Colorado, and Oklahoma. This knowledge can help shelter employees in these states better understand desirable phenotypes of dogs in their care.

This study demonstrates that imputed phenotypes such as skull type, predicted adult size, or coat length may be more predictive of outcome than breed or breed group alone. Previous studies have identified breed as a primary factor affecting shelter dog outcome, with some researchers finding that purebred dogs were more likely to be adopted than mixed breed dogs [10], that breed groups such as lap dogs, cocker spaniels, giant companion breeds, and “ratters” were more likely to be adopted compared to large companion breeds [8], and that toy, terrier, hound, and nonsporting breed groups were more adoptable than comparison groups [9]. Because the American Kennel Club (AKC) breed groups are phenotypically variable [34], it is often difficult to standardize phenotypes of such groups accurately and, therefore, it is likely that each shelter employee may identify the breed and purebred status of dogs entering the shelter differently. Unless genetic analyses are performed on all dogs entering each shelter, misidentification of breed may occur and subsequently lead to false results when analyzing the effect of breed type on adoption. Rather, a more accurate system for shelters to implement in their record keeping system may be through maintaining photographs of dogs when entering the shelter, in addition to recording phenotypic characteristics such as skull type, coat length, size, coat color, and/or coat pattern, and their best estimation of breed.

Misclassification bias may have been introduced during phenotype assignment from recorded breed. When imputing the skull type, estimated adult size, and coat length variables, there were some recorded breeds that were impossible to impute additional phenotypes from. For instance, some dogs were labelled as “terriers” or “mixed breed dogs” in which researchers were not able to impute phenotypes from such a general breed identification. Therefore, usable information from these dogs was limited. It is also possible that the actual appearance of each dog was not accurately described by the recorded breed, as it has been previously found that shelter workers may purposely mislabel dog breed if they reside in a community with breed specific legislation [35].

The results of this study provide an indication of the types of shelter dogs that adopters prefer. This evidence should allow shelter employees to make more objective decisions regarding the outcomes of dogs in their care by utilizing shelter record keeping to determine which dogs are preferred by adopters in their communities. Using the knowledge identified in this study, we speculate that shelter workers may be able to pick out certain apparently healthy dogs for adoption and forgo shelter health or behavioral protocols to reduce LOS, a shelter practice referred to as “fast tracking” [36]. This study suggests that shelter employees might attempt to “fast track” puppies and dogs small or medium in size but not dogs with a brachycephalic skull type, to expedite their chances of adoption and reduce their length of shelter stay. It is important to note that a system such as fast tracking is not applicable to every dog in the shelter, but to those that regardless of intake protocols, would be the first dogs to be adopted successfully. Fast tracking shelter dogs could potentially elevate the number of returned adoptions or promote the adoption of dogs without full adopter understanding of what each dog requires in terms of care. However, as the alternative to adoption is commonly euthanasia, the practice of fast tracking may be the best option for dogs in certain shelters.

An additional benefit of fast tracking may be that as dogs are removed from the adoptable shelter population, shelter employees will be able to properly allocate resources to dogs that may require a longer LOS, such as senior dogs or large dogs. These non-fast-tracked dogs could instead be sent onto transfer programs or remain in the shelter because their perceived adoptability may increase over time. Shelters could then properly allocate funds to dogs that previously would be utilized on housing dogs that are phenotypically more adoptable. Ultimately, this might allow shelters to increase their live release rate and reduce the number of dogs euthanized by freeing up kennel space from fast-tracked and transferred dogs.

## 5. Conclusions

This study supports the hypothesis that phenotypes in conjunction with length of shelter stay influence adopters’ preference of shelter dogs in the United States. Furthermore, the hazard for each phenotype for shelter dog adoption is not constant but is modified by the dogs’ length of shelter stay. This information is most directly applicable to municipally-funded shelters and may help shelter employees make more objective, phenotype-based outcome decisions for dogs entering their care.

## Figures and Tables

**Figure 1 animals-10-01959-f001:**
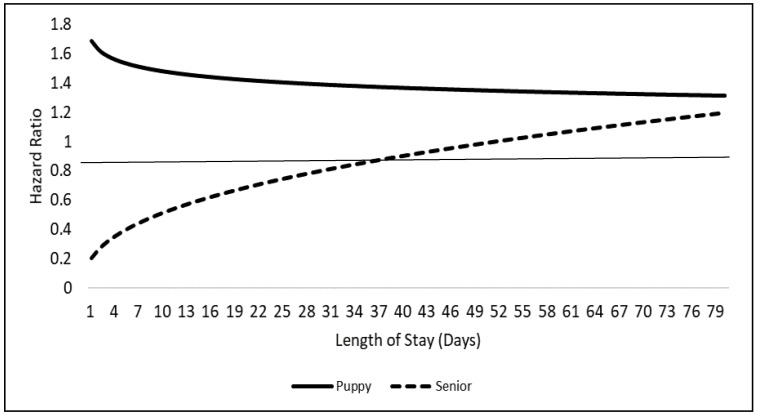
Model adjusted age group by length of stay interaction displayed as hazard ratios, estimated from the simple random sample of 4500 dogs, compared to the referent adult group.

**Figure 2 animals-10-01959-f002:**
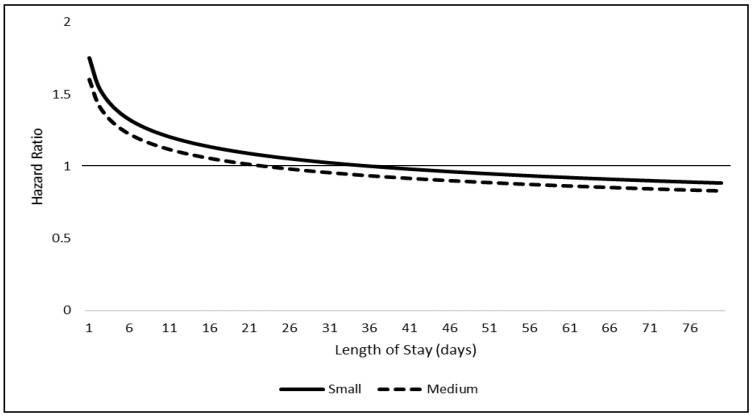
Model adjusted estimated adult size by length of stay interaction displayed as hazard ratios, estimated from the simple random sample of 4500 dogs compared to the referent large group.

**Figure 3 animals-10-01959-f003:**
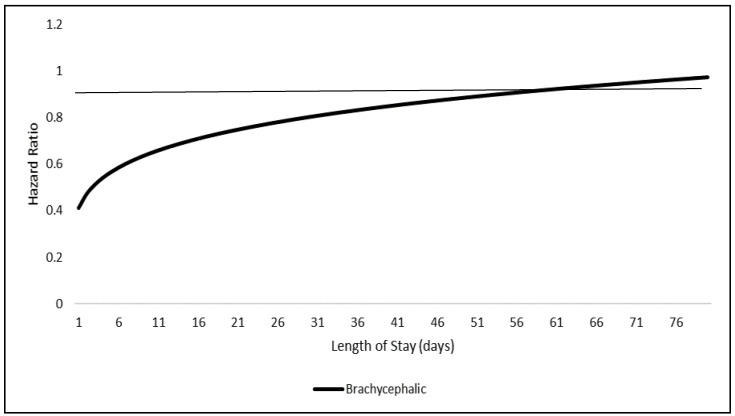
Model adjusted skull type by length of stay interaction displayed as hazard ratios, estimated from the simple random sample of 4500 dogs compared to the referent mesocephalic group.

**Table 1 animals-10-01959-t001:** Frequencies of phenotypes tested for multivariate adoption model inclusion using the full dataset 19,514 shelter dogs and the simple random sample (SRS) of 4500 dogs.

Variable	Response	Counts	Frequency (%)	Observations	SRS Counts	SRS Frequency (%)	SRS Observations
Coat Length	Short	13,214	69	19,287	3058	69	4451
	Long	6073	31		1393	31	
Skull Type	Brachycephalic	5875	32	18,648	1340	31	4324
	Mesocephalic	9162	49		2084	48	
	Dolichocephalic	3611	19		900	21	
Estimated Adult Size	Small	5503	28	19,356	1237	28	4467
	Medium	4290	22		986	22	
	Large	9563	49		2244	50	
Blockhead Type	Present	4163	21	19,514	976	22	4500
	Not Present	15,351	79		3524	78	
Coat Color	Black	5920	37	16,150	1370	36	3763
	Blue	364	2		95	2	
	Brown	2147	13		514	14	
	Grey	490	3		114	3	
	Red	1064	7		245	7	
	Tan	3304	20		749	20	
	White	2539	16		601	16	
	Yellow	322	2		75	2	
Gender	Male	10,020	52	19,302	2266	51	4443
	Female	9258	48		2177	49	
Age Group	Puppy	4026	22	18,605	911	21	4289
	Adult	12,973	69		2997	70	
	Senior	1606	9		381	9	
Region	South	3948	20	19,514	871	19	4500
	North	7168	37		1749	39	
	West	8398	43		1880	42	

**Table 2 animals-10-01959-t002:** Extended Cox regression model for adoption using the simple random sample of 4500 dogs including time interactions, indicated by length of stay (*LOS), for variables failing to meet model assumptions.

Parameter	ParameterEstimate	StandardError	Chi-Square	Pr > ChiSq	HazardRatio	95% Hazard Ratio Confidence Limits
Puppy	0.52299	0.12883	16.4794	<0.0001	1.687	1.311	2.172
Senior	−1.60083	0.23890	44.9008	<0.0001	0.202	0.126	0.322
Adult	referent						
Small	0.56154	0.12863	19.0589	<0.0001	1.753	1.363	2.256
Medium	0.47117	0.13638	11.9366	0.0006	1.602	1.226	2.093
Large	referent						
Brachycephalic	−0.88982	0.13745	41.9113	<0.0001	0.411	0.314	0.538
Dolichocephalic	−0.16450	0.13538	1.4764	0.2243	0.848	0.651	1.106
Mesocephalic	referent						
North	1.14019	0.38305	8.8604	0.0029	3.127	1.476	6.626
West	1.33720	0.41783	10.2423	0.0014	3.808	1.679	8.638
South	referent						
Puppy *LOS	−0.05625	0.05468	1.0583	0.3036	0.945	0.849	1.052
Senior *LOS	0.40597	0.08945	20.6003	<0.0001	1.501	1.259	1.788
Adult *LOS	referent						
Small *LOS	−0.15707	0.05284	8.8366	0.0030	0.855	0.771	0.948
Medium *LOS	−0.15029	0.05948	6.3844	0.0115	0.860	0.766	0.967
Large *LOS	referent						
Brachycephalic *LOS	0.19637	0.05461	12.9290	0.0003	1.217	1.093	1.354
Dolichocephalic *LOS	0.04106	0.05722	0.5150	0.4730	1.042	0.931	1.166
Mesocephalic *LOS	referent

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
