# Peer review of "Phenotypic Characteristics Associated with Shelter Dog Adoption in the United States"

_animals, 2020, doi:10.3390/ani10111959_

Round 1

Reviewer 1 Report

The situation in the US, i.e. to (over)use EUTHANASIA as a "containment" method, as you mentioned in your answer to my suggestions, in my PERSONAL opinion should find increasing opposition by humane scientists. I wonder if any final additions in the conclusions may even subtly touch such an ethical point. However, I leave the Authors total freedom to discard my present suggestion.

Author Response

Thank you for your suggestion. The authors feel that condemning or condoning euthanasia is outside the scope of this manuscript. Therefore, discussing the ethics of shelter dog euthanasia may be a more appropriate topic for a separate publication.

Reviewer 2 Report

Overall a good paper, however, I have a lingering question that one of the edits made stronger.   I would take further convincing that adopters are avoiding brachycephaly per see (a neotenic trait c.f. people preferring puppies), and not just avoiding pit bulls which are highly prevalent in many regions and highly stigmatized. This outcome is speculated on in the discussion when it could be largely answered via the data. Does this finding endure when pit pulls are excluded?

Author Response

Thank you for your suggestion. The authors agree that adopter preference away from brachycephalic dogs may be heavily influenced by the breed label of pit bull. However, because breed shelter records are highly variable and often unreliable, the authors would not have much confidence in the results of a test that excludes pit bull type breeds to determine brachycephalic adoptability. An explanation has been added to the discussion (Lines 273-275).

This manuscript is a resubmission of an earlier submission. The following is a list of the peer review reports and author responses from that submission.

Round 1

Reviewer 1 Report

    5 Baby schema in human and animal faces induces cuteness perception and gaze allocation in children. Borgi M, Cogliati-Dezza I, Brelsford V, Meints K, Cirulli F. Front Psychol. 2014 May 7;5:411. doi: 10.3389/fpsyg.2014.00411. eCollection 2014. PMID: 24847305 Free PMC article.                     6 Effectiveness of a Standardized Equine-Assisted Therapy Program for Children with Autism Spectrum Disorder. Borgi M, Loliva D, Cerino S, Chiarotti F, Venerosi A, Bramini M, Nonnis E, Marcelli M, Vinti C, De Santis C, Bisacco F, Fagerlie M, Frascarelli M, Cirulli F. J Autism Dev Disord. 2016 Jan;46(1):1-9. doi: 10.1007/s10803-015-2530-6. PMID: 26210515    

Nice paper, yet it is strangely limited to the US context. It should be definitely discussed and framed in a more extensive perspective.

    Those two papers, and the very important Lorenzian baby schema, need to be

an unaivoidable part of the Discussion and quoted:

Pet Face: Mechanisms Underlying Human-Animal Relationships. Borgi M, Cirulli F. Front Psychol. 2016 Mar 8;7:298. doi: 10.3389/fpsyg.2016.00298. eCollection 2016. PMID: 27014120 Free PMC article. Review   5 Baby schema in human and animal faces induces cuteness perception and gaze allocation in children. Borgi M, Cogliati-Dezza I, Brelsford V, Meints K, Cirulli F. Front Psychol. 2014 May 7;5:411. doi: 10.3389/fpsyg.2014.00411. eCollection 2014. PMID: 24847305 Free PMC article.   Specific points:   lines 99-104: the skull classification is absolutely simplistic and therefore unsufficient. Eg, in the methods, a minimal literature analysis of the analogous classification(s) for Homo sapiens and other Prime species could be advisable.  More importantly, the above-mentioned BABY SCHEMA shoul also find place in the dog classifiation, taking into account facial characteristics, eye position, etc.   191-195: again, such a result may be highli misleading, since it confuses skull type with general face expression attactiveness.   254-269: the present "geographical" classification does not allow to take into account sociologial and or cultural anthropological factors, varying fron State socioeconomic status to degree of recent immigration, etc, This is a very relevant component of the analytic part of both Discussion and Conclusion, which are definitely poor, if not scheletric, in the present MS.   293-297: The idea to arrange and proposing a scientifically-sound fat-track system is a very attractive one. However, mostly based on ethical issues, very relevant in the public opinion (and not necessarily embedded in a clear legislative act) in some European countries the track included rather geographycally and socioeconomically "distant" zones: in fact, eg big-sized dogs are much easily adopted in rural areas, while small-sized ones find an easy and confortable familial nucluus in inner cities, and also peri-urban zones with fragmented habitats showed quite different levels of acceptance than areas of skyscraper buildings.  This should be fully reported, not providing the false idea that (for dogs!) a fast-track destiny is a real good solution.

Author Response

Thank you for your suggestions. The authors included the paper by Borgi et al, 2014 in the Discussion to introduce the concept of baby schema (Lines 230-234).

Skull type was inferred from breed information found in the shelter records and, therefore  specific facial characteristics could not be implied from the data provided,  Although, the reviewer is concerned that skull type is too superficial, our analysis showed a relationship between skull type and outcome.  Therefore, skull type was useful and provided some valuable information.  Specific facial characteristics may also affect dog outcome, but this study does not have the resolution to determine cuteness on an individual level.

Next, regarding the geographic classification of shelters, the author’s intentions were to group shelter location based primarily on state but because some states only had one shelter providing information, it was impractical for us to generalize the effect of shelter dog adoption states in that situation as a whole. Rather a more general, regional association was created. With that said it would be inappropriate to look at societal factors affecting each region because there is such variability within each state, let alone each region of the US. A statement addressing this concern was added to the discussion (Lines 292-295). Thank you for the suggestion.

Lastly, to address fast tracking. Thank you for your suggestion. In overcrowded shelters in regions of the US with particularly elevated shelter dog overpopulation, currently the only outcomes for dogs are either adoption, euthanasia, or transfer. However, transfer programs are a relatively new concept to shelters and are therefore limited by shelter participation. Proposing fast tracking as a potential positive outcome for shelter dogs in areas of the country with increased pet overpopulation presents as a more favorable outcome for these dogs. The idea of fast-tracking is ideally meant to be interpreted as an alternative to euthanizing dogs by moving them through the sheltering system faster. It is also important to note that a system such as fast tracking is not applicable to every dog in the shelter, but to those that regardless of intake protocols, would be the first dogs to be adopted successfully. As you are suggesting, such dogs may include smaller sized dogs in urban areas or larger sized dogs in rural communities but would vary for each shelter dependent on shelter location. Therefore, if shelters begin to keep sufficient records and perform similar processes as outlined in this study, they could potentially increase dog adoption and decrease euthanasia. Performing similar methods on a shelter specific scale as opposed to a more generalized approach as demonstrated by this study. A sentence has been added to the discussion (Lines 334-339)

Reviewer 2 Report

Overall this is a well designed and presented study of interest to a wide audience.  I have the following suggestions.

1) Add at least one of the major findings to the abstract

2) Comment on how the method of imputation coded very high-frequency types (pit bull type, chihuahua) and the implications for their presence in a category.

3) Did sign in access to databases include any access to confidential data and if so how was this handled.

Author Response

Thank you for your suggestions and comments regarding our submission. Responses to the numbered reviews are listed below.

1) The major finding that lists the phenotypes that the study models has been added to the abstract (Lines 33-36).

2) As I understand the reviewer’s comment, the reviewer is asking to further explain how imputing phenotypes from breed assignment lead to certain phenotypes having a higher frequency of dogs and how this affected the resulting model. In response, an explanation was added to the materials and methods section to address this issue of falsely increasing or decreasing phenotype frequency from shelter misclassification of breed (Lines 108-116).

3) The data obtained from shelter databases were not considered confidential, as it was a report of yearly intakes and outcomes. Therefore, data confidentiality was not issue for this study. Further explanation of data confidentiality was added to the materials and methods section of the manuscript (Lines 84-85).

Reviewer 3 Report

The manuscript: “Phenotypic Characteristics Associated with Shelter Dog Adoption in the United States” raises a very important topic of animal homelessness and how to prevent it. Creating various types of models that can, to some extent, increase the chance of dogs to be adopted, is worth considering. It is a pity that shelters do not want to cooperate and only 20% responded to the offer of research cooperation.

The work deals with the subject of classification of dogs by some phenotypic traits instead by determining their breed. The authors' remark on the frequently incorrect classification of the dog breed by shelter employees (line 57) is justified.

Phenotypic traits taken into account in the assessment were described very precisely, including predicted adult size, coat length, and skull type in the chapter Dataset Variables (line 91).

Interesting results were obtained that region (line 175) was associated with adoption and met the proportionality assumption while age (line 179), predicted adult size (line 185) and skull type (191) did not meet the proportionality assumption.

It turns out that many factors affecting adoption are correlated with the time spent in the shelter. Such studies have probably not been done and the observations of volunteers from shelters in several European countries are quite the opposite, i.e. the longer a dog stays in a shelter the less its chance for adoption.

Very interesting results were also obtained from observations of age groups (line 215). The authors claim (line 222): that the longer senior dogs remain in the shelter, the more adoptable they become in relation to other age groups with the same LOS.

Observations carried out in Poland (Goleman et all, 2014) show that dogs from the senior group (over 8 years old) are the least adopted. This is most often caused by fear of an increased cost of living and veterinary care of older dogs.

Although the shelters lead campaigns promoting senior dogs there are no rescue groups that would adopt older dogs. The aforementioned studies also show that most adoptions are in the group of small dogs (3-10 kg) as well as puppies and young dogs (age group 3-12 months) (Goleman M., Drozd L., Karpiński M., Czyżowski P., 2014. Black dog syndrome in animal shelters, Med. Weter. 2014, 70 (2), 122-127).

The authors also note that brachycephalic dogs are always less adoptable than mesocephalic dogs and explain this fact that some of those brachycephalic breeds include dangerous dogs like: pit bulls, bulldogs, and rottweilers (line 248).

Howerver, another reason for little interest in adopting dogs of those breeds may be that e.g. small brachycephalic breeds are susceptible to many respiratory and cardiovascular diseases (for example: pug, french bulldog, king chales spaniel).

Many people make decision on adopting a dog by taking into account its phenotypic features. However, what I miss in this study, is a mental / behavioral aspect.

Anomalies in behavior and aggression are the most common causes of dogs being returned to shelters and their euthanasia.

It would be good if the authors  explained why psychological features were not included in the study (I realize that the reports may not have been mentioned).

However, in addition to the phenotypic features specified by the authors (as age, skull type, predicted adult size, or coat length), introduction of just the term "in the type of breed" could be an extension of this model, because the breed largely defines the behavior of a dog.

Training at least one shelter employee in dog breeds could solve this problem, as well as consultation with AKC members and breeders of purebred dogs. There is a danger that shelters will want to use "fast tracking" and give away dogs only on the basis of the phenotype, not taking into account the anomalies in their behavior. This why the risk of aggressive dog behavior should be assessed by using other protocols, as well.

The manuscript can be published in the journal Animals after minor revision.

Author Response

Thank you for your suggestions and comments. To begin, the study you referred to performed in Poland has been added to the discussion section of this paper (Lines 228-229). Additionally, the comment about increased prevalence of disease in brachycephalic dogs was added to the discussion section (Lines 273-275). The authors greatly appreciate these suggested additions.

Next, the authors do agree that behavioral characteristics influence shelter dog outcome, a finding that has been supported many times by previous researchers. However, the idea of this study was to determine that it’s not just behavior, but also phenotype that plays a part in whether shelter dogs are adopted or not. Additionally, behavioral information was vastly missing in the records provided, so such an analysis of the effect of behavior in combination with phenotype would not have been possible/accurate (very biased if we used the information we had). With that being said, an explanation has been added to the Materials and Methods section to acknowledge that breed would be the preferred variable analyzed, had it been a reliable variable (Lines 108-116).